# OpenReview forum: "Improving Tool Calling Accuracy for Large Language Models"
_ICLR.cc/2026/Conference — Submitted to ICLR 2026_

### Official Review · Reviewer_RBtx · 2025-10-20

**Soundness:** 1
**Presentation:** 1
**Contribution:** 1
**Rating:** 0
**Confidence:** 5

**Summary:**

Considering that the paper is clearly incomplete, I believe it should be strongly rejected.

**Strengths:**

Provides limited empirical comparison across models and datasets.

**Weaknesses:**

1. The paper is incomplete and lacks depth; it reads like an early-stage draft.
2. The idea—replacing structured JSON with a natural-language template—is trivial and does not constitute a substantive research contribution.

**Questions:**

None

---

> ### Author Response · Authors · 2025-11-14
>
> Dear Reviewer,
>
> Thank you for taking the time to read our paper and provide valuable feedback. As this is our first submission, we are eager to improve and learn from the review process.
>
> We would greatly appreciate it if you could provide more specific guidance on which parts of the paper gave the impression of being "clearly incomplete" or "like an early-stage draft." Understanding your concerns in detail will help us address them more effectively and strengthen the work.
>
> Thank you again for your time and insights.

---

### Official Review · Reviewer_H8Ad · 2025-10-22

**Soundness:** 3
**Presentation:** 1
**Contribution:** 1
**Rating:** 2
**Confidence:** 4

**Summary:**

This paper proposes a template-based approach to tool calling for LLMs: instead of emitting JSON or another schema-constrained format, the model produces natural-language templates which are then parsed into structured calls. Experiments on API-Bank, ToolACE, and When2Call with several models suggest gains on many F1 metrics.

**Strengths:**

Schema brittleness is a real source of failure in function calling. Demonstrating fewer schema violations via NL templates is a useful angle.

**Weaknesses:**

1. The core technique appears to be a prompt swap followed by regex parsing. There’s little discussion of design choices, alternative templating schemes, or robust parsing. Relative to the breadth of tooling literature, the contribution as framed feels incremental.
---
2. This paper’s writing quality is poor. It contains almost no discussion of related work, gives only a cursory treatment of the core method, devotes much of the main text to unstructured result listings with visible whitespace, and does not provide implementation details.
---
3.This paper asserts that templates align better with NL pretraining, but does not provide mechanistic or error-mode evidence beyond counts such as which violations decline.

**Questions:**

API-Bank L3 or ToolACE multi-call dialogs stress plan quality. Do templates help with tool selection across turns, or only with single-shot formatting?

---

> ### Author Response · Authors · 2025-11-14
>
> Dear Reviewer,
>
> Thank you for taking the time to read our paper and provide valuable feedback! Below, we address each point in detail and clarify any misunderstandings.
>
> 1. "The core technique appears to be a prompt swap followed by regex parsing. There's little discussion of design choices, alternative templating schemes, or robust parsing. Relative to the breadth of tooling literature, the contribution as framed feels incremental."
>
> We agree that exploring additional design choices — such as alternative prompts, templating schemes, and more robust parsing — would strengthen the study. However, as mentioned in the paper, our limited resources required us to prioritize breadth across models and datasets. Adding another dimension of variation, such as extensive prompt-design comparisons, was beyond our current capacity but remains a promising direction for future work.
>
> 2. "This paper's writing quality is poor. It contains almost no discussion of related work, gives only a cursory treatment of the core method, devotes much of the main text to unstructured result listings with visible whitespace, and does not provide implementation details."
>
> We acknowledge that the related work section is too brief. As this is our first submission, we mistakenly assumed readers would already be familiar with the background, which led to underspecification. We will substantially expand this discussion.
>
> Regarding the comment on "unstructured result listings with visible whitespace", we would greatly appreciate clarification on which specific tables or sections felt unstructured or contained visible whitespace issues. This feedback will help us revise the formatting more effectively.
>
> For "does not provide implementation details", could you specify which details you found missing? We included the full implementation code in the supplementary material to support reproducibility, but we may have overlooked necessary in-text explanations. Any additional guidance you can provide would be very valuable.
>
> 3. "This paper asserts that templates align better with NL pretraining, but does not provide mechanistic or error-mode evidence beyond counts such as which violations decline."
>
> As noted in the paper, we do not yet have a mechanistic explanation. Our contribution is based on the following process:
>  - We identified a plausible hypothesis about template alignment with training data.
>  - We proposed a novel approach informed by this hypothesis.
>  - We conducted experiments that aligned with our expectations.
>
> We acknowledge that deeper mechanistic or error-mode analysis would further strengthen the argument and consider this an important direction for future work.
>
> Regarding the question: "API-Bank L3 or ToolACE multi-call dialogs stress plan quality. Do templates help with tool selection across turns, or only with single-shot formatting?"
>
> This is indeed an interesting and important area. We have not yet evaluated templates in multi-turn or production-level complex schemas. Understanding how templates behave in multi-step planning, cross-turn tool selection, and real-world tool ecosystems is an exciting direction for future study.
>
> Thank you again for your time and insights. Your feedback is helping us significantly improve our work.

---

### Official Review · Reviewer_g2fS · 2025-10-28

**Soundness:** 2
**Presentation:** 1
**Contribution:** 2
**Rating:** 2
**Confidence:** 4

**Summary:**

This paper proposes a template-based generation method to improve LLM tool-calling accuracy, replacing conventional schema-constrained outputs with natural language templates. The approach is evaluated across three datasets and four LLMs . Results show consistent F1 improvements for tool names/parameters in most tests. The method leverages LLMs' natural language alignment, reducing schema violations and enhancing contextual understanding. However, performance varies with model architecture.

**Strengths:**

(1)Addresses a limitation of schema-constrained tool calling (misalignment with LLMs' natural language training).

(2)Template-based generation is simple yet effective, offering a practical alternative to rigid schemas.

(3)Tests diverse models and uses multiple datasets with varying complexity. Includes statistical significance testing.

**Weaknesses:**

(1)The paper argues that template-based generation is superior to schema-constrained generation, based on the assumption that the template-based format is more closely aligned with natural language. However, these assumptions and views lack both in-depth theoretical analysis and rigorous experimental verification.

(2)Calculating the Macro F1 score separately for the tool name and tool parameters in the experiment seems inappropriate, as incorrect names or incorrect parameters can lead to failure. Only correct names and parameters can lead to task success. Furthermore, the 0.9 threshold set for semantic similarity is arbitrary, and different embedding models can also affect this threshold.

(3)The paper's logic and presentation lack rigorousness, and many principles, concepts, and terminology lack prior explanation and clarification. In some areas, the description is incomplete and unclear.

**Questions:**

(1)Why is template-based generation better than schema-constrained generation? The paper concludes that "Current LLMs are predominantly pretrained and fine-tuned for schema-constrained generation (e.g., JSON output)."(line 383) Based on this, schema-constrained generation should be better.

(2)What's the rationale for setting the semantic similarity threshold at 0.9?

(3)Are there better metrics for measuring tool calling accuracy? The F1 score doesn't directly reflect the success or failure of tool calling.

(4) (Table 2 and Table 3) Why does GPT5 perform comparable to or better than other models on the API-Bank L1 and When2Call datasets, regardless of whether it's a schema-constrained or template-based method, but perform worse than almost all other models on the API-Bank L2 and ToolAce datasets?

(5)Why not compare with other tool calling studies? After all, simply relying on improvements to prompt format and style isn't necessarily an effective approach.

---

> ### Author Response · Authors · 2025-11-14
>
> Dear Reviewer,
>
> Thank you for taking the time to read our paper and provide valuable feedback! As this is our first submission, we are eager to improve and learn from the review process.
>
> We would greatly appreciate it if you could provide more details regarding which principles, concepts, or terminology "lack prior explanation and clarification" and which parts are "incomplete and unclear". A clearer understanding of these concerns will help us address them effectively and strengthen the work.
>
> Below, we respond to your specific questions:
>
> 1. "Why is template-based generation better than schema-constrained generation? The paper concludes that 'Current LLMs are predominantly pretrained and fine-tuned for schema-constrained generation (e.g., JSON output).' (line 383) Based on this, schema-constrained generation should be better."
>
> Natural language and JSON differ substantially. Although LLMs are optimized for generating JSON, the dominant pretraining data for most models remains natural language. This situation is analogous to comparing a task with limited training data but extensive optimization against a task with abundant data but minimal optimization—neither is guaranteed to be superior.
> Our experiments demonstrate that, even without dedicated optimization, template-based generation can achieve comparable or better results than schema-constrained generation.
>
> 2. "What's the rationale for setting the semantic similarity threshold at 0.9?"
>
> This threshold is based on empirical experience, and we acknowledge it is not strictly rigorous. We welcome your thoughts on whether comparing raw strings directly — without semantic similarity — would be preferable.
>
> 3. "Are there better metrics for measuring tool-calling accuracy? The F1 score doesn't directly reflect the success or failure of tool calling."
>
> This is an excellent point. Due to limited resources, we focused primarily on the schema generation aspect and assumed a perfect downstream environment — i.e., tool execution always succeeds when provided with correct tool names and parameters. Exploring more direct tool-success metrics is a valuable direction for future work.
>
> 4. "(Table 2 and Table 3) Why does GPT5 perform comparable to or better than other models on the API-Bank L1 and When2Call datasets, regardless of whether it's a schema-constrained or template-based method, but perform worse than almost all other models on the API-Bank L2 and ToolAce datasets?"
>
> Unfortunately, we do not yet have a concrete explanation for this behavior.
>
> 5. "Why not compare with other tool-calling studies? After all, simply relying on improvements to prompt format and style isn't necessarily an effective approach."
>
> We agree that comparing with other tool-calling approaches would be valuable. However, given our limited resources, we focused on strengthening our study through a broader combination of models and datasets. Adding another dimension of comparison — different methodological approaches — was beyond our current capacity but remains promising future work.
>
> Thank you again for your time and insights. Your feedback is helping us significantly improve our work.

---

### Official Review · Reviewer_6R8D · 2025-11-01

**Soundness:** 1
**Presentation:** 1
**Contribution:** 1
**Rating:** 0
**Confidence:** 4

**Summary:**

This paper proposes to generate tool calls from models by using a natural language template rather than a json-like schema. It is shown to be better than using a json-like schema in some scenarios.

**Strengths:**

N/A.

**Weaknesses:**

1. Lack of comparison with chat templates that models are trained in. LLMs nowadays are trained to use tools following a certain template (see https://huggingface.co/docs/transformers/main/en/chat_extras). This can either be achieved using chat templates for open source LLMs or using tool call APIs for closed LLMs (https://platform.openai.com/docs/guides/function-calling). Since models are trained this way, it is only fair to compare with them in this setting, instead of directly putting the tool schemas in the prompt.

2. Lack of comparison with constrained decoding methods. Lots of the errors spotted by the authors such as schema violation, incorrect tool name, incorrect parameter name, can be avoided with constrained decoding, because it limits the vocabulary to only the valid tokens during a tool call. Constrained decoding/structured output has been implemented by major closed LLMs (https://platform.openai.com/docs/guides/structured-outputs) and open-source inference engines such as sglang (https://docs.sglang.ai/advanced_features/structured_outputs.html). Hence, a major part of the problem that this paper is solving no longer exists.

3. Marginal improvements over the baseline. In Table 2 and Table 3, many scenarios show worse results for template-based tool calling than schema-based. I'm not sure if this method is only working for certain scenarios, as there is no motivation or justification behind its effectiveness.

**Questions:**

see weakness.

---

> ### Author Response · Authors · 2025-11-14
>
> Dear Reviewer,
>
> Thank you for taking the time to read our paper and provide thoughtful feedback! Below, we address each point in detail and clarify any misunderstandings.
>
> **1. Concern regarding chat templates**
>
> We did experiment with chat templates and included notes in the submitted code supplement explaining why we did not report results using them. We acknowledge that these notes should have been included in the paper itself. The main reasons are:
> - Mistral’s chat template has limitations regarding the order of prompts, which does not fit datasets like API-Bank.
> - DeepSeek-Coder fine-tuned with chat templates produces empty outputs on datasets like When2Call, suggesting that special handling may be required when fine-tuning this model on the dataset.
>
> To maintain consistency across different models and datasets, we chose not to use chat templates in the reported results.
>
> **2. Concern regarding constrained decoding (CD)**
>
> Constrained decoding improves accuracy, but it is a general technique applicable to both schema-based and template-based methods. Currently, most ecosystems are heavily optimized for JSON-based methods, and APIs such as those from OpenAI do not support regex-based constrained decoding.  Therefore, evaluating constrained decoding would create an unfair advantage for schema-based methods—not due to inherent technical superiority, but because of limitations in the current ecosystem.
>
> **3. Concern regarding marginal improvements**
>
> The models with limited performance are DeepSeek-Coder and GPT5.
> - DeepSeek-Coder is primarily trained on code, which falls outside our assumption that models are trained mostly on natural language. Its lower performance was expected and serves as a worst-case illustration.
> - GPT5: Section 4.5 shows that reasoning impacts performance. While its task optimizations are not publicly known, our results suggest it is not well-suited for template-based methods, making its observed performance consistent with expectations.
>
> We appreciate the reviewer’s thoughtful comments, which allowed us to clarify these points. Overall, our results show that template-based methods remain competitive, even when considering the ecosystem’s inherent bias toward schema-based approaches. We hope these clarifications address the concerns raised and provide a clearer understanding of our contributions.

---

### Meta-Review · Area_Chair_Dpk6 · 2026-01-06

**Summary:**

The reviewers consistently raised serious concerns about the novelty, rigor, and completeness of the submission, and is presented with insufficient methodological detail, weak theoretical grounding, and poor writing quality, all of which substantially weaken the case for a meaningful contribution.

**Reviewer Concerns:**

No concerns are addressed by the rebuttal.

**Reviewer Scores:**

No scores are changed.

---

### Decision · Program_Chairs · 2026-01-26

Reject